# Weak acid and pepsin reflux induce laryngopharyngeal mucosal barrier injury: A rabbit-model-based study

Chenjie Hou[1,2,3], Liqun Zhou[4], Yujin Zheng[5], Ting Chen[1,2,3]*, Renyou Hu[6], Jingyi Zheng[7], Chaofeng Liu[8], Yaqi Liu[1,2,3]

1 Department of Otorhinolaryngology, Fujian Provincial Hospital, Fuzhou, Fujian, China, 2 Department of Otorhinolaryngology, Shengli Clinical Medical College of Fujian Medical University, Fuzhou, Fujian, China, 3 Department of Otorhinolaryngology, Fuzhou University Affiliated Provincial Hospital, Fuzhou, Fujian, China, 4 Department of Otorhinolaryngology, Shishi General Hospital, Quanzhou, Fujian, China, 5 Department of Otorhinolaryngology, Zhongshan Hospital, Fudan University (Xiamen Branch), Xiamen, Fujian, China, 6 Chongqing Jinshan Science & Technology (Group) Co Ltd, Chongqing, China, 7 Department of Otorhinolaryngology, Fujian Provincial Maternity and Children's Hospital, Fuzhou, Fujian, China, 8 Department of Otorhinolaryngology, Fujian Provincial Governmental Hospital, Fuzhou, China

☯ These authors contributed equally to this work.
* iambbp@vip.sina.com

**Data Availability Statement:** The laboratory protocols is deposited in protocols.io (DOI: dx.doi.org/10.17504/protocols.io.eq2lywy8rvx9/v1). All

## Abstract

### Objective

Using rabbit models, this study simulated the laryngopharynx's response to the synergistic effects of various acidic reflux environments and pepsin to investigate the response mechanism underlying weak acid reflux and pepsin in the mucosal barrier injury of laryngopharyngeal reflux.

### Methods

The rabbits were divided into six groups, and the original larynx was recorded for each group. During the study period, rabbits were sprayed with different doses of acid and pepsin solutions and monitored for hypopharyngeal mucosal transient impedance before and after modeling. After the experiment, laryngeal mucosal tissues were collected, observed using hematoxylin and eosin staining, and assessed for E-cadherin expression. The width of the intercellular space and lanthanum staining penetrating the intercellular space were also observed using electron microscopy.

### Results

Eight weeks post-modeling, evidence of laryngopharyngeal mucosa inflammatory responses was observed in each group. Downregulation of E-cadherin expression significantly positively correlated with acid strength (p < 0.05). The pepsin and acid intervention groups showed a significantly widened space between mucosal epithelial cells in the posterior ring area (p < 0.05). Meanwhile, in the experimental group, a large amount of stained

relevant data are within the manuscript and its Supporting information files.

**Funding:** All three of the funders — Major Scientific Research Program for Young and Middle-aged Health Professionals of Fujian Province, China (Grant no. 2022ZQNZD001); Health youth research project of Fujian Province (Grant no. 2022QNH006) and Leading Project Science and Technology Innovation Joint Fund of Fujian Province (Grant no. 2023Y9351) — had no role in study design, data collection and analysis, decision to publish, or preparation of the manuscript.

**Competing interests:** The authors have declared that no competing interests exist.

lanthanum penetrated the intercellular spaces; however, no significant difference was observed in the mucosal impedance (MI).

## Conclusion

This study demonstrated that acid, weak acid, and pepsin could damage the laryngeal mucosal barrier; pepsin was an independent factor associated with tissue damage; the downregulation of hypopharyngeal cadherin was associated with acid-intensity exposure. Transient LP-MI cannot be applied directly.

## 1. Introduction

Laryngopharyngeal reflux disease (LPRD) is an inflammatory disease condition resulting from gastroduodenal reflux, leading to morphological changes in the upper airways and digestive tract [1]. LPRD has been linked to the occurrence of various severe laryngeal diseases [2]; however, the pathophysiology of the injury in this disease remains unclear. Weak acids are believed to play significant roles in LPRD [3, 4]. We have previously reported the potential involvement of non-acidic reflux in laryngopharyngeal reflux [5]. In addition, pepsin plays an important role in the pathogenesis of LPRD [6]. Compared with that of the esophagus, the mucosa of the throat lacks a resistance mechanism to gastric acid and, hence, is more susceptible to reflux injury. Acid and pepsin connect perturbation decomposition of the complex barrier function of throat mucosa epithelial permeability and damage.

However, existing research mainly focuses on the damage caused by strong acids to the laryngeal mucosa. Current treatment methods for LPRD primarily target laryngeal reflux induced by strong acids, while the effectiveness against weak acid and non-acid (pepsin) reflux is relatively poor [7, 8]. This difference is related to the distinct mechanisms of injury that weak acids and pepsin exert on the larynx compared to the direct damage caused by strong acids. Therefore, this study aims to investigate the effects of weak acids and pepsin on laryngeal injury. E-cadherin is essential to maintaining epithelial integrity by mediating homogenous adhesion at junctions where epithelial cells attach. In a study involving 18 patients with LPRD, Gill et al. [9] found that E-cadherin expression in the laryngeal mucosal epithelium was significantly reduced, causing the destruction of the related intercellular barrier and the loss of mucosal continuity.

Notably, genetic diversity significantly contributes to the variations of calcium mucin expression and its underlying protection mechanisms. Therefore, diagnosing LPRD solely by pH monitoring may be unreliable, which necessitates novel diagnostic indicators that better reflect mucosal injury.

Mucosal impedance (MI) is determined by the intrinsic conductivity of the mucosa. In *In vitro* animal experiments [10], mucosal impedance can reflect the status and integrity of the mucosa, offering a new diagnostic index for reflux pharyngitis. This study simulated the laryngopharynx's response to the synergistic effects of various acidic reflux environments and pepsin to investigate the response mechanism underlying weak acid reflux and pepsin in the mucosal barrier injury of laryngopharyngeal reflux.

## 2. Materials and methods

The study was approved by the Animal Ethics Committee of Fujian Medical University (No.: FJMU IACUC).

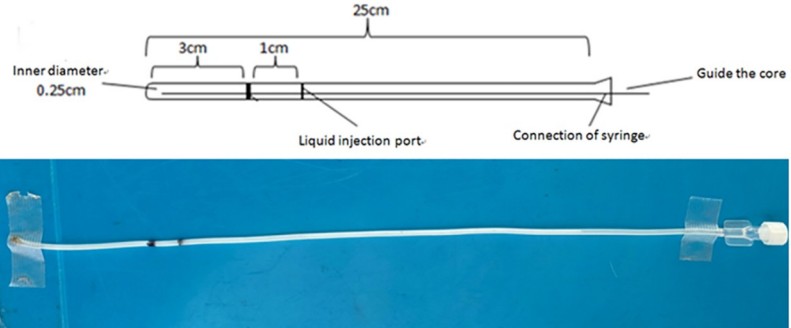

**Fig 1. Schematic diagram of the catheter.**

## 2.1. Animal

Thirty healthy male New Zealand albino rabbits with corporal mass between 2.5 and 3.5 kg were used in this study. To induce laryngitis, the rabbits received intramuscular anesthesia with diazepam (3 mg/kg; Xudong Haipu Pharma Co., Ltd., Shanghai, China), ketamine (40 mg/kg; Gutian Pharma Co., Ltd., Fujian, China), and Su-Mian-Xin (4 mg/kg; Shengda Pharma Co., Jilin, China). A special catheter (Fig 1) was inserted through the nasopharynx until it reached the hypopharynx. The external tip was sutured to the nasal lateral cartilage.

The rabbits were randomly assigned into six groups. During the 8-week study, each group was treated with pH 3, 5, 7, and normal saline solution as the matrix (Table 1), with or without pepsin at 1 mL/kg for 3 min of slow spraying, twice per day, through the special catheter.

All rabbits underwent endoscopic observation with a 3-mm endoscope (70° for nose, STORZ, Germany) and LP-MI detection with a transient Impedance-pH Reflux Monitoring Systems (Chongqing Jinshan Science & Technology (Group) Co., Ltd.) immediately and 8 weeks after intubation. At the end of the intubation period (8 weeks), the rabbits were euthanized, and the larynx and pharynx were removed for morphological and histological analysis.

## 3. Experimental method

### 3.1 Morphological evaluation: Laryngeal morphology assessment under laryngoscope

Following intramuscular anesthesia, a laryngoscopy was conducted. Since assessment of normal values for rabbits is lacking, we referred to the methods of Hu et al. and Lou et al. [11, 12], using a standard reflux finding score (RFS) system described by Belafsky et al. [13] to analyze the images. The RFS was calculated by two independent observers.

### 3.2 Instantaneous impedance monitoring of hypopharyngeal mucosa

After being anesthetized and fixed, the hypopharynx of all rabbits was monitored using a transient MI measurement system (Chongqing Kingsoft Technology Co., Ltd., China) (Fig 2)

**Table 1. Groups of experimental animals by intervention.**

|  | Acid (pH = 3) | Weak acid (pH = 5) | Non-acid |
|---|---|---|---|
| With pepsin | Experiment Group 1: Acid + Pepsin (n = 4) | Experiment Group 2: Weak acid + Pepsin (n = 5) | Experiment Group 3: Normal saline + Pepsin (n = 5) |
| Without pepsin | Experiment Group 4: Acid (n = 5) | Experiment Group 5: Weak acid (n = 5) | Control group: Normal saline (n = 6) |

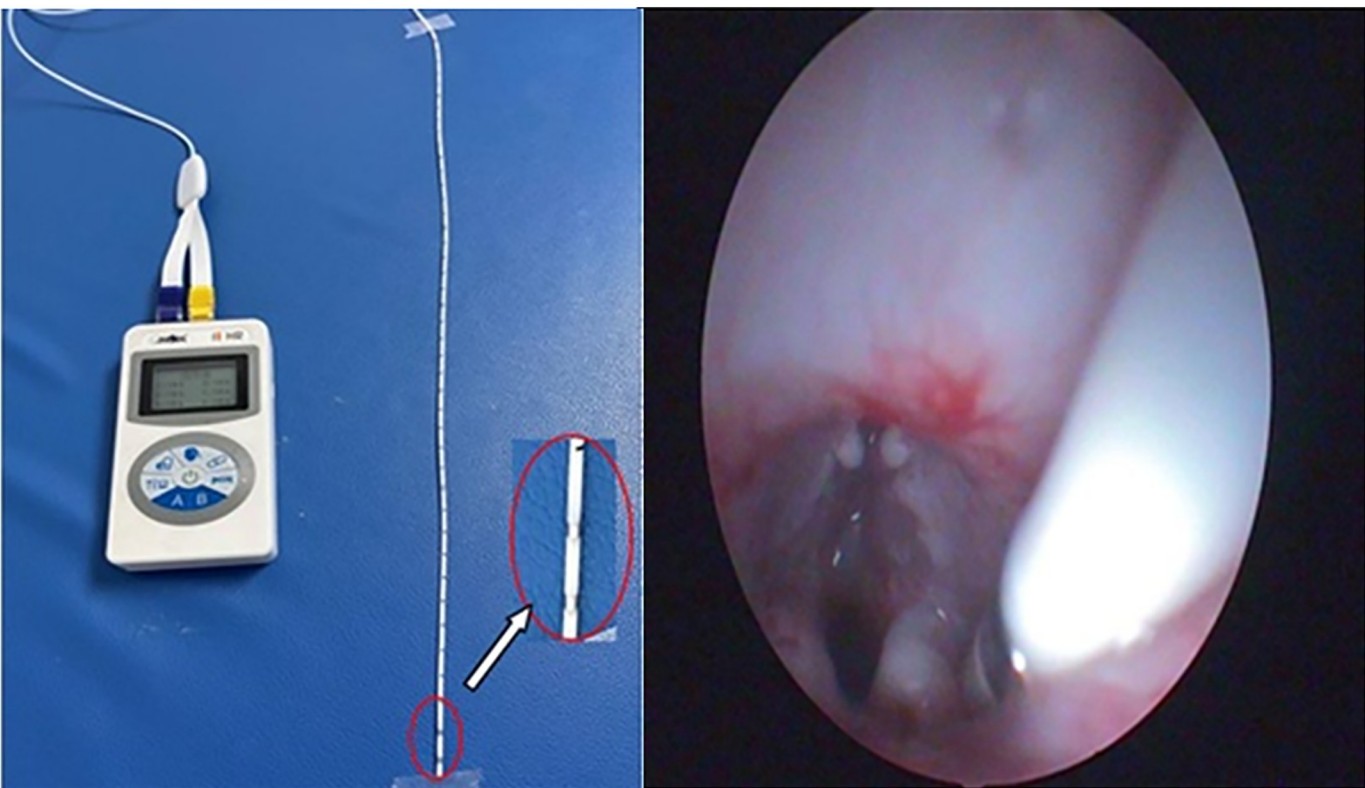

**Fig 2. Schematic diagram of impedance measurement.**

before and after catheter placement and removal. Data from experimental animals in a calm breathing state were recorded for 5 min.

### 3.3 Histopathology

After 8 weeks, the rabbits were sacrificed using air embolism. The mucosal tissue on the inner side of the arytenoid area (non-catheterized side) was pruned into small tissue blocks of approximately $1 \times 1 \times 1$ mm3 on an ice tray and examined using electron microscopy to calculate after lanthanum nitrate staining, dilated intercellular spaces (DIS), hematoxylin and eosin staining, and immunohistochemical examination.

### 3.4. Statistical analysis

Stata MP 13 statistical software package was employed for analyses. Normally distributed data are shown as mean ± standard deviation, whereas abnormally distributed data as median (interquartile range). Parametric differences were compared using the *t*-test, while nonparametric differences using the Kruskal—Wallis test. The correlation was compared using Spearman correlation analysis. p value < 0.05 was considered statistically significant.

## 4. Results

### 4.1. General information

No significant difference was identified in body weight between each group before and 4 and 8 weeks after modeling (p > 0.05).

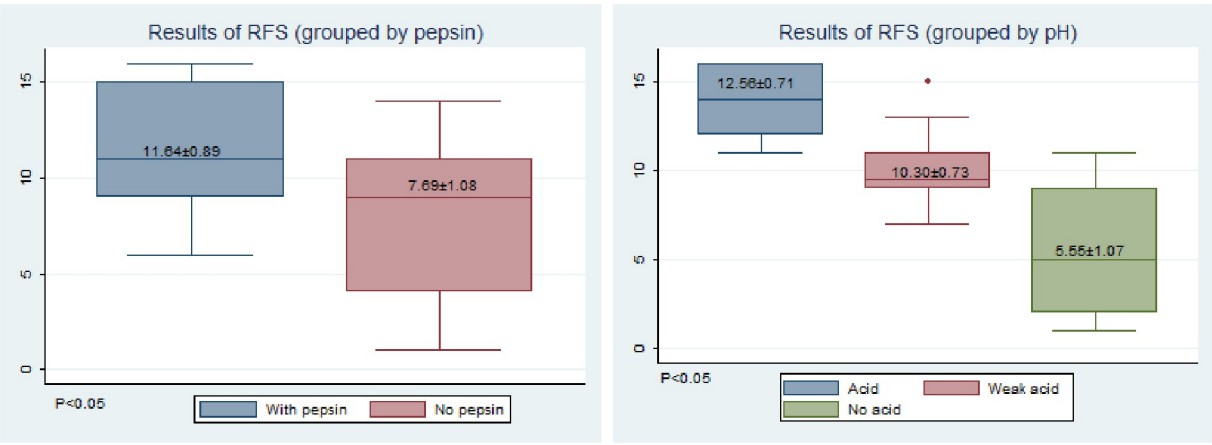

**Fig 3. Results of RFS.** (A)Grouped by pepsin. (B) Grouped by pH.

## 4.2. Morphological evaluation under laryngoscope

The vocal cords of New Zealand rabbits are composed of non-keratinized laminated squamous epithelium. Before modeling, no significant differences were observed in the total RFS scores of each group ($p > 0.05$).

The total RFS scores were higher at 8 weeks post-modeling than before modeling. The mean score of the pepsin exposure group was significantly higher than that of the non-pepsin exposure group (Fig 3a). According to pH groups, the average score was positively correlated with acid exposure, and the difference between groups was significant (Fig 3b).

Significant differences were observed among the scores of false vocal fold sulcus, vocal fold edema, and diffuse laryngeal edema, regardless of whether in the pepsin or acid exposure group, and in the scores of laryngeal ventricular disappearance and posterior combined hyperplasia among the groups with different pH values.

## 4.3. Hypopharyngeal transient MI monitoring analysis

The implanted probe missed the arytenoid area. The mean MI values before and 8 weeks after modeling showed no significant differences ($1,903.03 \pm 132.23$ and $2,050.83 \pm 175.93$, respectively) ($p = 0.50$). Moreover, no significant difference was found among the MIs of the groups ($p = 0.52$). However, the mucus in the larynx significantly reduced the impedance results.

## 5. Transmission electron microscope evaluation

### 5.1 Lanthanum dyeing observation

Transmission electron microscopy revealed that the space between the mucosal epithelial cells in the posterior ring area was significantly widened in all experimental groups, and a large amount of stained lanthanum passed through the intercellular space. The acid + pepsin group showed necrosis and lanthanum staining. In the saline control group, osmic acid staining did not, or very rarely, pass through the intercellular space; no cell necrosis was identified (Fig 4).

### 5.2 Intercellular space measurement

The mean intercellular space of the pepsin exposure group was significantly higher than that of the non-pepsin exposure group (DIS) (Fig 5a). In the pH groups, the intercellular space was

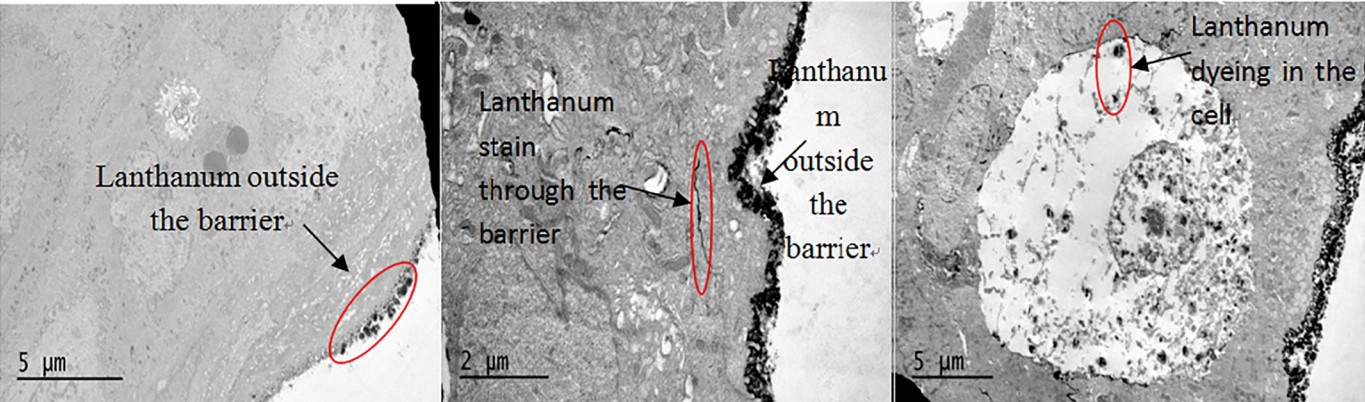

**Fig 4. Transmission electron microscope lanthanum staining diagram.** (A)Negative: lanthanum staining did not penetrate the barrier (control group). (B) Lanthanum staining penetrated the cellular barrier (experimental group). (C)Cell necrosis, lanthanum penetrated the cell (only in acid + pepsin experimental group).

positively correlated with acid exposure, and the difference between the groups was significant (Figs 5b and 6).

## 6. Inflammatory observation

Under a light microscope, unlike the normal saline control group, the experimental groups showed partial necrosis and shedding of the epithelial layer of the posterior ring area, infiltration of lymphocytes and plasma cells in the lamina propria, rupture of the muscular layer, hyperplasia and hypertrophy of the submucosal glands, vasodilatation, and congestion (Fig 7).

The inflammatory response was positively correlated with the exposure intensity, whether in the pepsin or acid group, and the differences between the groups were significant ($p < 0.05$).

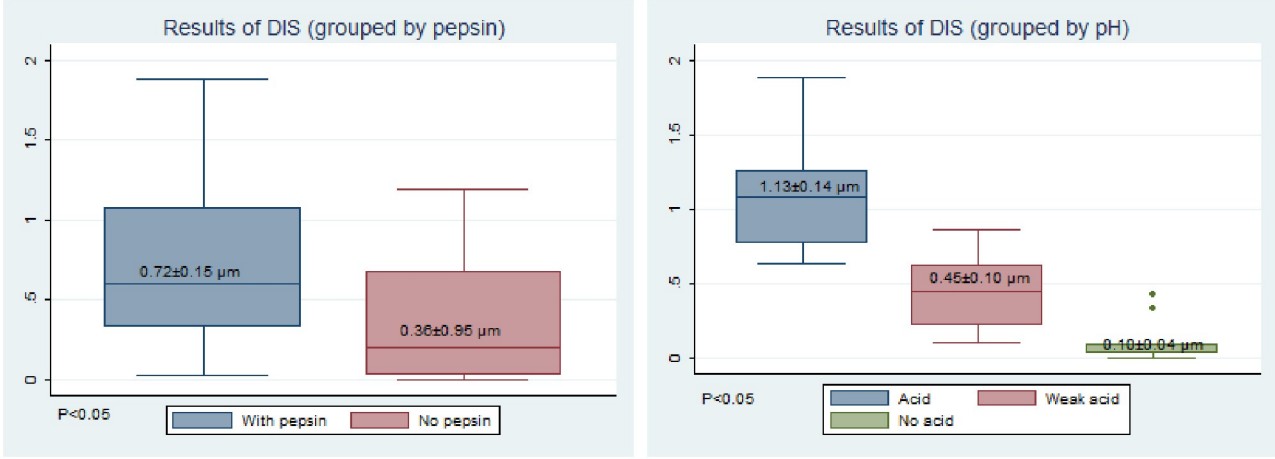

**Fig 5. Comparison of the DIS results.** (A)Grouped by pepsin. (B) Grouped by pH.

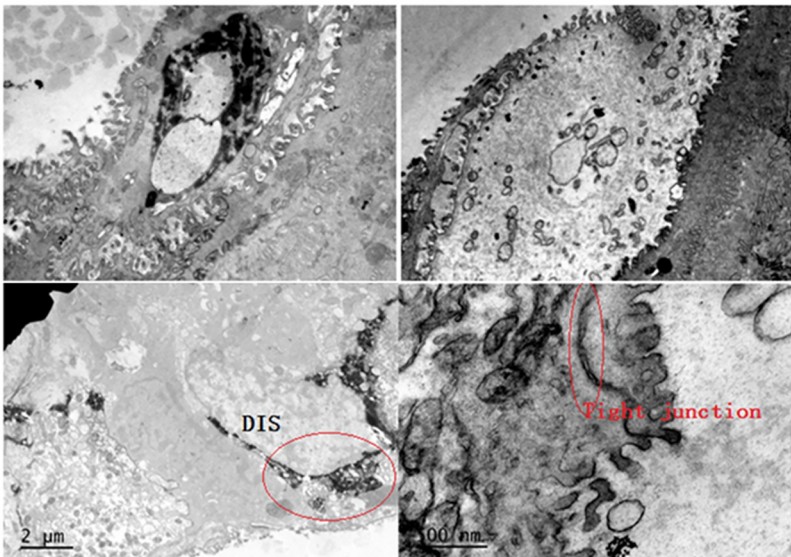

**Fig 6. Observation of transmission electron microscopy (conventional staining).** (A) Apoptotic vacuoles with lysosomes. (B) Cell edema c. Expansion of intercellular space d. Cells are closely connected.

## 7. E-cadherin expression

Spectrophotometry after immunohistochemical staining showed that the downregulation of E-cadherin was positively correlated with the acid exposure intensity (Fig 8). There was no significant difference when assessing pepsin exposure (Fig 9a). Data from each group were tested using the Shapiro—Wilk test; the p-values were > 0.05, indicating a normal distribution. Significant differences were observed among the three groups (Fig 9b).

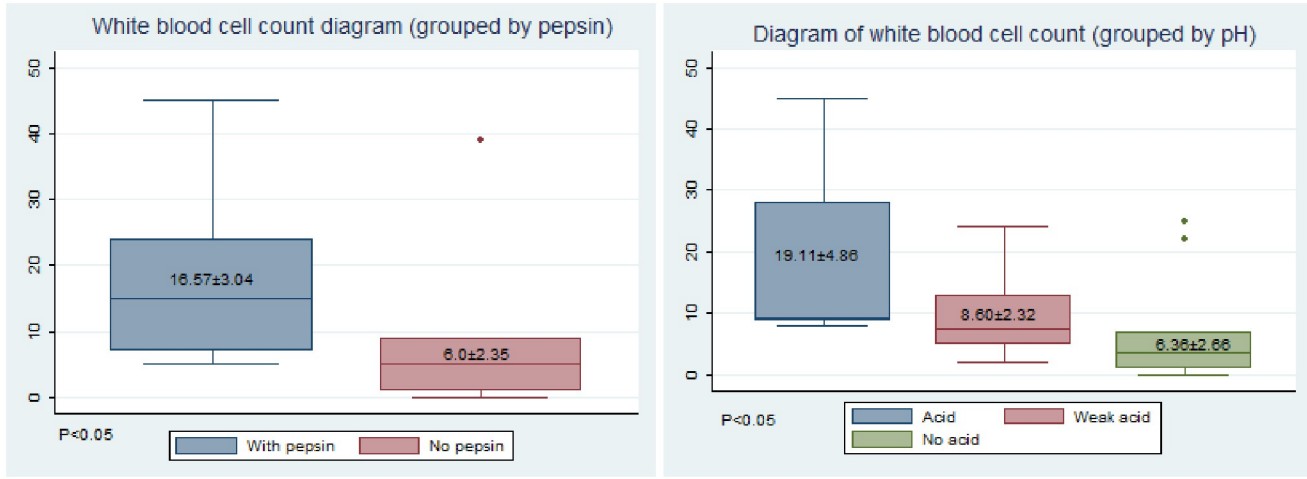

**Fig 7. Degree of inflammation (white blood cell count diagram).** (A)Grouped by pepsin. (B) Grouped by pH.

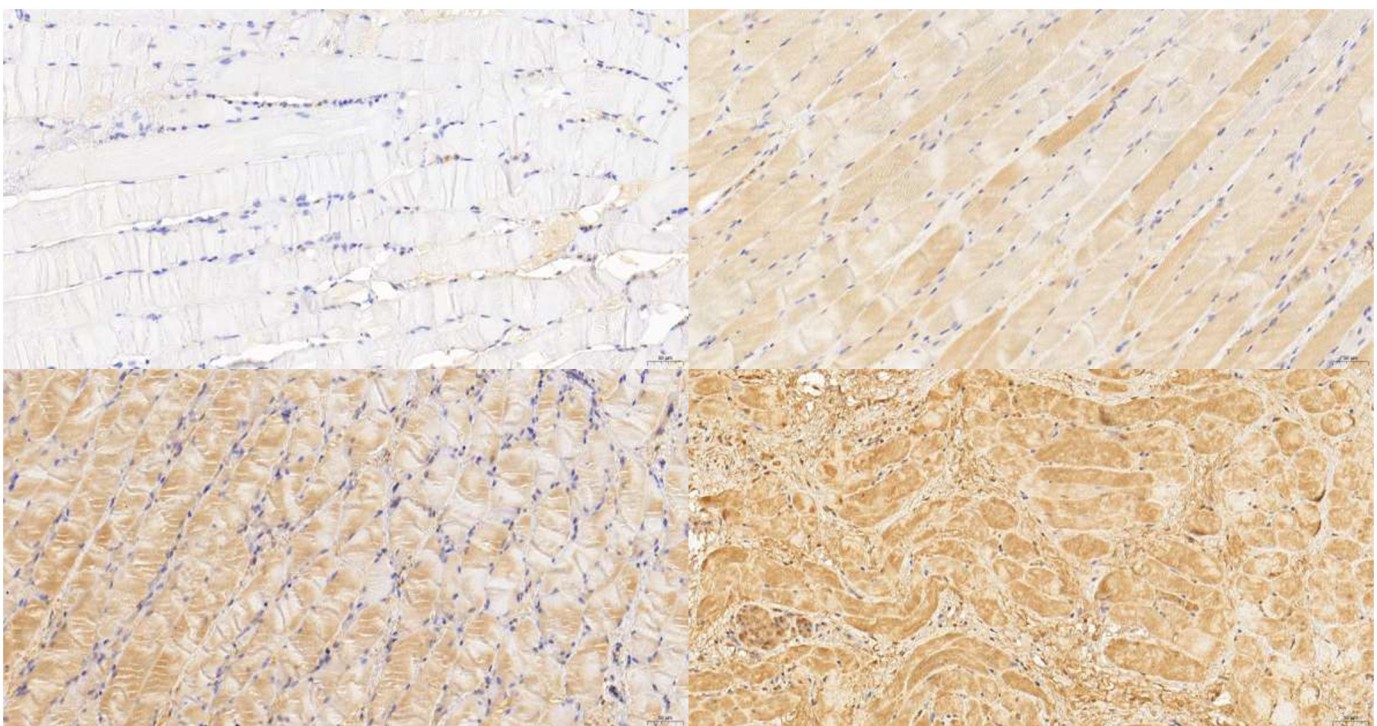

**Fig 8. Intensity grading of E-cadherin immunohistochemical staining using spectrophotometry.** (A) Level 0 (B) Level 1 (C) Level 2 (D) Level 3.

## 8. Discussion

The study of weak acid, specifically non-acid, on laryngopharyngeal damage has attracted attention. In a prospective study of 99 patients, Palareti [14] found that glottic edema was positively and significantly associated with the number of non-acid LPR and non-acid esophageal reflux events. Laryngeal compartment disappearance and posterior junction hyperplasia

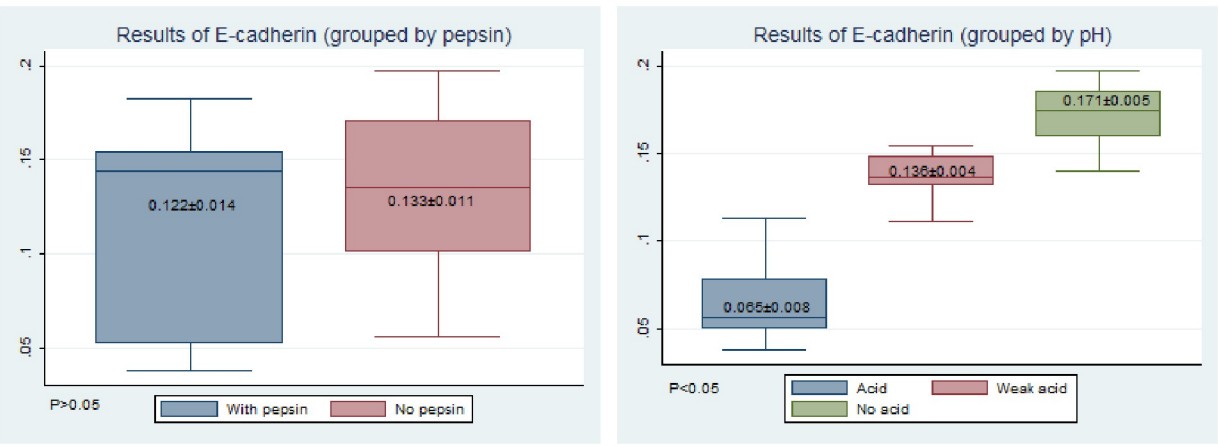

**Fig 9. Results of E-CAD staining compared by spectrophotometry.** (A)Grouped by pepsin. (B)Grouped by pH.

significantly correlated with non-acidic and total reflux exposure times. A significant correlation was also found between granuloma/granulation score and the number of acidic LPR events.

In a study of 349 patients, Li et al. [15] found that non-acid reflux events play a considerable role in LPRD; non-acid is a potential risk factor for laryngeal inflammation, consistent with the results of this study. A considerable number of patients do not respond well to antacid therapy, which may be related to the poor effect of drugs on weak acid and non-acid reflux.

Existing studies suggest that pepsin is a primary component of laryngopharyngeal reflux injury, which induces mucosal inflammation by damaging the mucosa and is a specific and sensitive biomarker of LPR. Pepsin can be found in the laryngeal mucus or laryngeal mucosal cells of patients with reflux and is involved in the pathogenesis of vocal cord polyps, vocal cord leukoplakia, and laryngeal cancer [16].

Inactive but stable pepsin (pH 7 in non-acidic reflux) can be actively absorbed by laryngeal epithelial cells in a receptor-mediated manner via acid-independent endocytosis [17]. Pepsin can also be retained and reactivated in some intracellular vesicles [18], leading to intracellular damage and changes in pro-inflammatory cytokine gene expression. Johnston et al. [19] confirmed that endorsed pepsin causes mitochondrial damage and induces the expression of some stress and toxin-related genes. This suggests the involvement of pepsin in the signs and symptoms of weak acid and non-acid reflux. Samuels et al. [20] further investigated cell damage caused by pepsin and found that the gene expression profile of pro-inflammatory cytokines induced by endocytosed non-acidic pepsin was similar to that of severe gastroesophageal reflux disease (GERD).

Our study found that pepsin exposure independently resulted in significant laryngeal changes and severe inflammatory cell infiltration. In this study, the endocytosis of pepsin, independent of gastric acid, explains the observation of cell barrier destruction and morphological changes under laryngoscopy, even in the non-acid group (water + pepsin group). At the same time, our study established the animal model of weak acid and non-acid reflux for the first time and realized the effect of pepsin on the throat in the animal model without acid.

The downregulation of E-cadherin has been observed in the laryngeal mucosa of patients with LPRD [21], which causes laryngopharyngeal mucosa injury and discomfort. In our study, E-cadherin downregulation was also observed, and the degree of downregulation was negatively correlated with pH. E-cadherin, also known as epithelial cadherin or CD324, is a calcium-dependent cell adhesion molecule. Decreased E-cadherin expression is a marker of epithelial-mesenchymal transition and associated with an increased risk of cancer metastasis. The expression level of E-cadherin is correlated with the degree of tumor differentiation of head and neck squamous carcinoma and negatively correlated with the metastasis of head and neck carcinoma and patient prognosis [22]. A higher incidence of laryngeal and hypopharyngeal cancers has been reported in patients with laryngopharyngeal reflux; therefore, its relation to the decrease or loss of E-cadherin expression caused by reflux warrants further studies.

Samuels et al. [23] have demonstrated that exposure to pepsin only for a short duration could activate cancer-related signaling pathways in laryngeal cells. Li et al. [24] found that acidizing pepsin promoted metabolic reprogramming from oxidative phosphorylation to aerobic glycolysis by reducing the activity of mitochondrial respiratory complex I and enhanced the growth and migration ability of vocal cord white spot epithelial cells. Active treatment of LPRD aims not only to improve the quality of life and relieve discomfort but also prevent and reduce the occurrence of hypopharyngeal and laryngeal cancer.

Simultaneously, the acid + pepsin group had the most serious mucosal damage, suggesting the expansion of the intercellular space, mucosal barrier damage, and apoptosis, consistent with previous findings.

Furthermore, obvious staining of lanthanum through the mucosal barrier in the weak acid and non-acid + pepsin experimental groups was also observed in this study, indicating that acid and non-acid reflux can induce expansion of the intercellular space and mucosal damage. The synergistic effect of pepsin cannot be ignored.

The esophageal mucosa of patients with GERD has ultrastructural changes, such as DIS [16]. The widening of the intercellular space is an early morphological marker of tissue damage in patients with GERD. Caviglia et al. [25] compared the intercellular spaces in the esophageal epithelia of patients with non-erosive esophagitis with those of normal individuals and concluded that a widened intercellular space is characteristic of patients with non-erosive esophagitis and can be used as an objective indicator of changes in the barrier structure.

Ravelli et al. [26] identified acid exposure as a major factor leading to space dilatation in the esophageal epithelium, and that the use of proton pump inhibitors reduces DIS, suggesting DIS as a feature of GERD. What's more, laryngopharyngeal reflux has some similarities to GERD etiology.

Li et al. [27] used transmission electron microscopy to show that with the onset of erosive esophagitis, mucosal thickness increased, desmosomes significantly decreased, and the space between epithelial cells widened. However, this measurement is difficult to apply in clinical practice since it depends on a mucosal biopsy under an electron microscope. The breakdown of the mucosal barrier is stable during the disease state and is not susceptible to interference from other factors [10]. When no gas or fluid passes through, the lumen of the esophagus collapses, and the esophageal mucosa contacts the metal ring of the catheter, resulting in MI, which is determined by the inherent electrical conductivity of the esophageal mucosa. Therefore, MI may be a novel, reliable, highly sensitive, and specific diagnostic index for reflux pharyngitis and can reflect the integrity of the esophageal mucosa. A lower MI value indicates more severe damage to the esophageal mucosal integrity, which is negatively correlated with DIS. However, LPRD affects pharyngeal mucosal epithelial space enlargement.

In LPRD, this study found that DIS was proportional to the acid intensity. In addition, the stability of the laryngeal instantaneous MI was poor, and no significant difference between the two groups before and after modeling was observed. Possible reasons are as follows: laryngeal and esophageal anatomies differ, the detection mode of laryngeal MI cannot be directly applied to the detection mode of esophageal MI, and its placement and detection method requires further studies and optimization. However, several factors strongly affect MI detection. For example, we found that with more secretions in the larynx, MI could be significantly reduced. Breathing and swallowing can also affect MI measurements. Therefore, in future studies, the timing of MI testing should be optimized. For example, patients could gargle to reduce the secretion of the pharyngeal cavity and avoid swallowing to improve the test accuracy.

## 9. Conclusion

This study found that acid, weak acid, and pepsin could damage the laryngeal mucosal barrier and identified pepsin as an independent factor that may cause damage, as manifested by the downregulation of hypopharyngeal cadherin and enlargement of DIS. Downregulation of hypopharyngeal cadherin is related to acid-intensity exposure. However, transient throat MI cannot be directly applied unless the influencing factors are removed.

## Supporting information

**S1 Dataset.**
(XLSX)

## Acknowledgments

We would like to express our gratitude to Editage (www.editage.cn) for English language editing services. We also thank Dr. Linying Zhou, Dr. Minxia Wu, and Dr. Xi Lin from the Electron Microscopy Lab at the Public Technology Service Center, Fujian Medical University, for their generous technical assistance with electron microscopy.

## Author Contributions

**Data curation:** Liqun Zhou, Jingyi Zheng.

**Investigation:** Yujin Zheng, Chaofeng Liu, Yaqi Liu.

**Methodology:** Renyou Hu.

**Writing – original draft:** Chenjie Hou.

**Writing – review & editing:** Ting Chen.

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
