## [Decision Letter · Decision Letter 0]

25 Jun 2024

PONE-D-24-20203weak acid and pepsin reflux  induced laryngopharyngeal mucosal barrier injury:a study  based on rabbit modelPLOS ONE

Dear Dr. Hou,

Thank you for submitting your manuscript to PLOS ONE. After careful consideration, we feel that it has merit but does not fully meet PLOS ONE’s publication criteria as it currently stands. Therefore, we invite you to submit a revised version of the manuscript that addresses the points raised during the review process.

We look forward to receiving your revised manuscript.

Kind regards,

Mahmoud Kandeel

Academic Editor

PLOS ONE

2. As part of your revision, please complete and submit a copy of the Full ARRIVE 2.0 Guidelines checklist, a document that aims to improve experimental reporting and reproducibility of animal studies for purposes of post-publication data analysis and reproducibility: https://arriveguidelines.org/sites/arrive/files/documents/Author%20Checklist%20-%20Full.pdf Please include your completed checklist as a Supporting Information file. Note that if your paper is accepted for publication, this checklist will be published as part of your article.

“Major Scientific Research Program for Young and Middle-aged Health Professionals of Fujian Province, China (Grant no. 2022ZQNZD001) and Health youth research project in Fujian Province (Grant no. 2022QNH006).”

Reviewers' comments:

Reviewer's Responses to Questions

**Comments to the Author**

1. Is the manuscript technically sound, and do the data support the conclusions?

Reviewer #1: Partly

Reviewer #2: Partly

2. Has the statistical analysis been performed appropriately and rigorously? 

Reviewer #1: Yes

Reviewer #2: Yes

3. Have the authors made all data underlying the findings in their manuscript fully available?

Reviewer #1: Yes

Reviewer #2: No

4. Is the manuscript presented in an intelligible fashion and written in standard English?

Reviewer #1: Yes

Reviewer #2: No

5. Review Comments to the Author

Reviewer #1: Dear Authors,

I have carefully reviewed your manuscript titled "Response of the Laryngopharynx to Acidic Reflux and Pepsin in Rabbit Models" and find it to be a promising and insightful study presenting novel findings on the effects of pepsin on laryngopharyngeal reflux. However, I believe the manuscript would benefit significantly from the following considerations:

1- Enhancement of English Punctuation and Style:

It would be advantageous to improve the punctuation and overall writing style to enhance readability and ensure the manuscript meets the highest standards of academic writing.

Grammar and Style:

A thorough proofreading to address minor grammatical errors and improve the overall readability of the manuscript.

2- E-Cadherin Expression:

- The expression of E-cadherin alone may not be a definitive indicator of cancer progression. Downregulation of this marker could also result from the wound healing process, considering the tissue damage duration.

The images provided for grading and scoring E-cadherin expression are of low resolution and should be improved. Furthermore, the images depicting high expression levels show a concerning degree of background staining.

- The significance of E-cadherin expression depends on both the intensity and localization of the staining within the tissue, particularly at cell-cell contacts. The image showing the lowest expression still indicates presence at cell-cell contacts, suggesting functional E-cadherin. Therefore, the assertion that this expression is downregulated and indicative of cancer progression is not entirely substantiated by the current images. I recommend including more images and additional markers to provide a clearer and more comprehensive analysis.

3- Consider evaluating additional markers related to cancer development or gene mutations, as they could provide more conclusive evidence.

4- Inflammatory Markers:

The inclusion of inflammatory markers and corresponding tissue images would serve as strong evidence of an inflammatory response. Also, addition of images of the tissue showing inflammation, tissue damage or any other findings would be beneficial.

5- Effect of Catheter and Standardization:

The potential impact of the catheter on the results should be addressed, and it is crucial to standardize the methodology to ensure that procedural variations do not influence the outcomes. Otherwise, providing explanation of the measures taken to reduce variability in results would provide more clarity.

6- Treatment Dosage:

The manuscript mentions that the tissue was sprayed with the test solution, but it lacks details regarding the exact volume used and whether the dosage was standardized across subjects. Providing this information is essential to avoid variability in treatment dosage.

7- Expand on Pepsin's Role:

Elaborate on the potential mechanisms of pepsin-induced damage in non-acidic conditions and its clinical implications.

8- Improve Figure Legends:

Ensure that all figure legends are detailed and fully explain the data presented, enhancing the reader's understanding.

While your research is promising and offers valuable insights, addressing these points will significantly enhance the manuscript's clarity, robustness, and overall impact. I encourage you to incorporate these suggestions to provide a more comprehensive and rigorous presentation of your findings.

Best regards,

Reviewer #2: I would like to thank the authors for their effort in this study. There are many serious concerns regarding this manuscript as highlighted below:

Title

I do not see what is special about this topic, “Weak acid and pepsin reflux induced laryngopharyngeal mucosal barrier injury: a study based on rabbit model Study LPRD on rabbit model.”

1. Weak acid is generally believed to be gastric acid. Why are the authors avoiding writing the proper term? They should be specific! The weak acid could be anything.

2. It is established knowledge that LPRD causes reflux. What is so special about this study? The significance of the study is not properly documented.

3. The mention of “Study LPRD on rabbit model” is unnecessary.

Abstract

The method is not detailed! Sprayed with different concentrations of acid— which acid? Hydrochloric or sulfuric?

The tissues were observed with Hematoxylin and eosin. For what? What was the observer looking for? This is not informative at all!

The conclusion part is not understood and rather confuses the readers, especially the statement “This study found that acid, weak acid, and pepsin…”

Introduction

Why is mentioning weak acid instead of gastric acid important in this context?

The two statements here, “Compared with the mucosa of the esophagus, the mucosa of the throat lacks a resistance mechanism to gastric acid and is more susceptible to reflux injury. Acid and pepsin disrupt barrier function and epithelial permeability by perturbing junctional complexes,” lack references.

The statement “Gill et al. [6] studied 18 patients diagnosed with LPRD and found that the expression of E-cadherin in the laryngeal mucosal epithelium was significantly reduced, and pepsin appeared” is not understood with the ending words “pepsin appeared.”

The following sentence should have been broken into two for clarity.

The statement “Notably, as a result of genetic diversity, individual expression of calcium mucin protection mechanisms must be differences” is poorly written and not scientifically constructed.

The sentence “Therefore, the existence of throat mucosa damage is defined, and the diagnosis of LPRD is unreliable” is not clear. Who defined it, or what does the author mean here?

The next statement, “Therefore, we may need to look for objective diagnostic indicators that can reflect mucosal injury,” is poorly written. Also, the knowledge gap here is different from what was stated in the abstract: “In this study, rabbit models were used to simulate the response of the laryngopharynx under the synergistic effects of various acidic reflux environments and pepsin to reveal the response mechanism of weak acid reflux…”

Therefore, the authors sound inconsistent with their objectives!

Methodology

The methodology was poorly described. For instance, the statement “The animals were randomly assigned into six groups (five animals per group). During the 8-week study, the rabbits in each group were treated with pH 3, 5, 7, and normal saline solution as the matrix, adding or not adding pepsin at 1 mL/kg for 3 min of slow spraying, twice per day, through the special catheter” is unclear.

Which group received what? This was poorly described. No one understands what was done to the animals and how each group was treated.

Also, the phrase in this statement “each group were treated with pH 3, 5, 7” is ambiguous and confusing.

In the statement “After intramuscular anesthesia, the animals received laryngoscopy. Because there was no previous assessment of normal values for rabbits, we referenced the method reported by Zhang et al. and Lou et al.[8,9] using a standard RFS system described by Belafsky et al.[10] to analyze images,” what does this mean? The author did not provide a detailed explanation of what those references mean. Are they not referring to rabbits? Are they pointing to abnormal values?

Also, the authors just jumped to “What is instantaneous impedance?” They failed to provide some important background information in the introductory section. They just jumped to the methodology without detailed information. Overall, the authors sound non-communicating and unscientific in their writing form.

The statement “After 8 weeks, the animals were sacrificed by air embolism…” Why air embolism? What does this suggest? There was no detailed explanation of how this impacts the results.

Also, what does lanthanum nitrate do? What is DIS? What do the immunostainers reveal, and what are those immunomarkers under the section “2.3 HISTOPATHOLOGY”? The information provided is below expectations without any details.

And so on….

Overall, the manuscript lacks clarity, proper documentation, and scientific rigor in its writing. The authors need to provide detailed explanations, proper references, and clear descriptions of their methodology and findings to make the study comprehensible and valuable.

6. PLOS authors have the option to publish the peer review history of their article (what does this mean?). If published, this will include your full peer review and any attached files.

Reviewer #1: **Yes: **Arwa Flemban

Reviewer #2: **Yes: **Kamoru Adedokun

---

## [Author Response · Author response to Decision Letter 0]

22 Aug 2024

Response to Reviewers

Reviewer #1

Dear Dr. Arwa Flemban,

Thank you very much for your detailed and constructive feedback on our manuscript entitled " Weak acid and pepsin reflux induce laryngopharyngeal mucosal barrier injury: a rabbit-model-based study". We appreciate the time and effort you have dedicated to evaluating our work, here is my response.

1- Enhancement of English Punctuation and Style:

The manuscript has been revised by the professional editing service: Editage (www.edita ge.cn).

2- E-Cadherin Expression:

Regarding E-Cadherin expression, we have updated the images with higher resolution and included references to E-Cadherin and cancer-related progression in the discussion section.

3- Consider evaluating additional markers related to cancer development or gene mutations, as they could provide more conclusive evidence.

We have also supplemented the discussion with relevant literature evidence on the link between inflammation and carcinogenesis. Due to the limited availability of rabbit-related antibodies, we are planning to search for more suitable markers for our future research.

4- Inflammatory Markers:

Added discussion on the inflammatory response section. 

5- Effect of Catheter and Standardization:

To ensure the reliability of our experimental results, we used a blank control group (normal saline group) to account for any influence of the catheter. Additionally, we selected non-catheterized tissues to minimize any potential effects.

6- Treatment Dosage:

In our study, we applied different concentrations of solvents to treat 1 mL/kg of rabbit throat for 3 minutes using slow spraying. 

7- Expand on Pepsin's Role:

We have discussed the mechanism of pepsin on cell damage and its potential impact on the development of laryngeal carcinoma.

8- Improve Figure Legends:

we have made modifications to part of the legend.

Best regards,

Reviewer #2

Dear Dr. Kamoru Adedokun,

Thank you for your insightful comments and suggestions on our manuscript titled " Weak acid and pepsin reflux induce laryngopharyngeal mucosal barrier injury: a rabbit-model-based study". We appreciate the time and effort you have invested in providing feedback. Below is my response.

1. Title

This study focuses on examining the effect of weak acid and pepsin on the laryngeal injury. Existing research has focused more on damage to the throat mucosa caused by strong acids. PPI treatment for laryngeal reflux effects the reflux caused by strong acid, while the treatment effect of weak acid and acid-free (pepsin) reflux is poor, which is related to the difference between the mechanism of damage caused by weak acid and pepsin to the throat and the mechanism of direct damage caused by strong acid. Therefore, this study focused on the effects of weak acid and pepsin on laryngeal injury. Common animal models include pigs, dogs, rabbits, and rats. The vocal cord structure of rabbits is the same as that of humans, and rabbits are silent animals, which can avoid the influence of vocal cord movement injury on the results. Therefore, we selected rabbits for modeling.

2. Abstract

Since the main component of stomach acid is hydrochloric acid, to simulate stomach acid, hydrochloric acid was used in this study. Hematoxylin and eosin were used to observe the tissue inflammatory response. The expression of some sentences was modified.

3. Introduction

Revised the wording in this section and added additional references.

4. Methodology

Since there is no specific rabbit laryngoscopy scoring standard, we also used the human laryngoscopy scoring standard previously reported (Belafsky et al.). Air embolization is necessary for rabbit tissue sampling and does not affect the experimental results. Lanthanum nitrate staining is one of the special staining methods for electron microscopy. When the mucosal barrier is intact, lanthanum nitrate does not penetrate the intercellular space; however, if the mucosal barrier is damaged, lanthanum staining can penetrate the intercellular space and inside the cell when the cell dies. DIS is the expansion of the cell gap; that is, when the mucosal barrier is normally intact, the cell connection is tight; however when broken, the cell gap can be observed under the electron microscope. Finally, the expression of E-Cadherin in cells was observed using immunohistochemical staining of arytenoid mucosa. Some background information regarding instantaneous impedance was added to the revised. And new references have been added.

Best regards,

---

## [Decision Letter · Decision Letter 1]

11 Sep 2024

PONE-D-24-20203R1Weak acid and pepsin reflux induce laryngopharyngeal mucosal barrier injury: a rabbit-model-based studyPLOS ONE

Dear Dr. Chen,

Thank you for submitting your manuscript to PLOS ONE. After careful consideration, we feel that it has merit but does not fully meet PLOS ONE’s publication criteria as it currently stands. Therefore, we invite you to submit a revised version of the manuscript that addresses the points raised during the review process.

We look forward to receiving your revised manuscript.

Kind regards,

Miquel Vall-llosera Camps

Senior Staff Editor

PLOS ONE

Journal Requirements:

Reviewers' comments:

Reviewer's Responses to Questions

**Comments to the Author**

1. If the authors have adequately addressed your comments raised in a previous round of review and you feel that this manuscript is now acceptable for publication, you may indicate that here to bypass the “Comments to the Author” section, enter your conflict of interest statement in the “Confidential to Editor” section, and submit your "Accept" recommendation.

Reviewer #1: All comments have been addressed

Reviewer #2: All comments have been addressed

2. Is the manuscript technically sound, and do the data support the conclusions?

Reviewer #1: Yes

Reviewer #2: Yes

3. Has the statistical analysis been performed appropriately and rigorously? 

Reviewer #1: I Don't Know

Reviewer #2: Yes

4. Have the authors made all data underlying the findings in their manuscript fully available?

Reviewer #1: Yes

Reviewer #2: (No Response)

5. Is the manuscript presented in an intelligible fashion and written in standard English?

Reviewer #1: Yes

Reviewer #2: Yes

6. Review Comments to the Author

Reviewer #1: Dear Author,

I am writing to provide feedback on your manuscript, "[Manuscript Title]". I have carefully reviewed the original manuscript and the amendments you have provided.

I am pleased to note that many of the original comments have been addressed satisfactorily in the revised manuscript. The improvements you have made have significantly enhanced the clarity and overall quality of the work.

However, I would like to express some lingering concerns regarding the images of the E-cadherin staining. Based on my experience, E-cadherin, as an active molecule, is typically localized at the cell-cell adhesion points. While the images provided demonstrate two distinct intensity levels of the marker, neither image clearly shows staining at the cell membrane. The staining appears more like background noise.

I would recommend repeating the E-cadherin staining experiment with a different antibody or considering an alternative marker to assess EMT status or cellular epithelial integrity. Here are some suggestions for potential markers:

Occludin: A tight junction protein that is often downregulated during EMT.

ZO-1: Another tight junction protein that can be used to assess epithelial integrity.

Vimentin: An intermediate filament protein that is upregulated during EMT.

I believe that addressing these concerns will further strengthen the manuscript and provide a more compelling representation of your findings.

Sincerely,

Reviewer #2: Thank you for explaining the concept. However, I find it concerning that your response to critiques, as well as those from other investigators I have come across in similar roles, often shows more clarity and strength than the original document. I notice that your revised explanation demonstrates greater energy and efficiency.

Could you incorporate this idea into your response to Reviewer 2, particularly regarding what your study contributes to existing knowledge on weak acids and why the treatment is important? You briefly touched on this in your response, but I believe this key information should have been highlighted in the abstract or introduction to provide a stronger foundation for the study.

It's essential for investigators to present their best insights upfront to engage and motivate readers.

Secondly, I would encourage you to pay close attention to capitalization in your title. It’s important to handle these details yourself, rather than relying on the journal office to perfect what should be done correctly from the start, especially after acceptance.

Good luck!

7. PLOS authors have the option to publish the peer review history of their article (what does this mean?). If published, this will include your full peer review and any attached files.

Reviewer #1: **Yes: **Arwa Flemban

Reviewer #2: **Yes: **Kamoru A. Adedokun

---

## [Author Response · Author response to Decision Letter 1]

10 Oct 2024

Response to Reviewers

Subject: Response to Reviewer #1’s Comments on Manuscript "Weak Acid and Pepsin Reflux Induce Laryngopharyngeal Mucosal Barrier Injury: A Rabbit-Model-Based Study"

Dear Dr. Flemban,

Thank you very much for your thoughtful suggestions and constructive feedback on our manuscript, "Weak Acid and Pepsin Reflux Induce Laryngopharyngeal Mucosal Barrier Injury: A Rabbit-Model-Based Study." We truly appreciate the time and effort you have dedicated to reviewing our work.

In response to your insightful comments regarding the E-cadherin staining images, we have chosen to upload new images that we believe more accurately reflect our experimental results.

We greatly value your recommendations on utilizing markers such as Occludin, ZO-1, and Vimentin in our future studies. E-cadherin, as you pointed out, is a specific adhesion molecule for epithelial cells, and its downregulation indeed signifies a loss of epithelial characteristics, thereby providing a clear indication of EMT occurrence. We also acknowledge that Occludin, ZO-1, and Vimentin can offer complementary insights that would enhance our understanding of the EMT process. Unfortunately, due to time constraints, we were unable to conduct additional experiments at this stage. We are committed to considering your suggestions in our ongoing research.

Thank you once again for your invaluable input and guidance.

Warm regards,

Ting Chen

Subject: Response to Reviewer #2’s Comments on Manuscript "Weak Acid and Pepsin Reflux Induce Laryngopharyngeal Mucosal Barrier Injury: A Rabbit-Model-Based Study"

Dear Dr. Adedokun,

Thank you very much for your valuable feedback and constructive suggestions regarding our manuscript, "Weak Acid and Pepsin Reflux Induce Laryngopharyngeal Mucosal Barrier Injury: A Rabbit-Model-Based Study."

We appreciate your observation about the clarity and strength of our response compared to the original document. In light of your feedback, we have revised the introduction to more clearly highlight the contribution of our study to the existing knowledge on weak acids and their implications for treatment. We have incorporated additional information and relevant references to provide a stronger foundation and underscore the importance of our findings.

Regarding the title capitalization, we have ensured that it adheres to the proper formatting guidelines. We understand the importance of these details and have taken care to address them appropriately in the revised manuscript.

Thank you once again for your insightful comments, which have significantly contributed to improving our manuscript.

Sincerely,

Ting Chen

---

## [Decision Letter · Decision Letter 2]

21 Nov 2024

Weak acid and pepsin reflux induce laryngopharyngeal mucosal barrier injury: a rabbit-model-based study

PONE-D-24-20203R2

Dear Dr. Chen,

We’re pleased to inform you that your manuscript has been judged scientifically suitable for publication and will be formally accepted for publication once it meets all outstanding technical requirements.

Kind regards,

Miquel Vall-llosera Camps

Senior Staff Editor

PLOS ONE

Reviewers' comments:

Reviewer's Responses to Questions

**Comments to the Author**

1. If the authors have adequately addressed your comments raised in a previous round of review and you feel that this manuscript is now acceptable for publication, you may indicate that here to bypass the “Comments to the Author” section, enter your conflict of interest statement in the “Confidential to Editor” section, and submit your "Accept" recommendation.

Reviewer #2: All comments have been addressed

2. Is the manuscript technically sound, and do the data support the conclusions?

Reviewer #2: Partly

3. Has the statistical analysis been performed appropriately and rigorously? 

Reviewer #2: Yes

4. Have the authors made all data underlying the findings in their manuscript fully available?

Reviewer #2: Yes

5. Is the manuscript presented in an intelligible fashion and written in standard English?

Reviewer #2: Yes

6. Review Comments to the Author

Reviewer #2: I have no other comments. I think the reviewers have satisfied with the critiques. Best of luck.

#########################################

#########################################

7. PLOS authors have the option to publish the peer review history of their article (what does this mean?). If published, this will include your full peer review and any attached files.

Reviewer #2: **Yes: **Kamoru Adedokun

---

## [Editor Report · Acceptance letter]

25 Nov 2024

PONE-D-24-20203R2 

PLOS ONE

Dear Dr. Chen, 

I'm pleased to inform you that your manuscript has been deemed suitable for publication in PLOS ONE. Congratulations! Your manuscript is now being handed over to our production team.

Kind regards, 

on behalf of

Dr. Miquel Vall-llosera Camps 

Staff Editor

PLOS ONE